# Co-Benefits of Energy Structure Transformation and Pollution Control for Air Quality and Public Health until 2050 in Guangdong, China

**DOI:** 10.3390/ijerph192214965

**Published:** 2022-11-14

**Authors:** Haihua Mo, Kejun Jiang, Peng Wang, Min Shao, Xuemei Wang

**Affiliations:** 1Guangdong-Hongkong-Macau Joint Laboratory of Collaborative Innovation for Environmental Quality, Institute for Environmental and Climate Research, Jinan University, Guangzhou 510632, China; 2Energy Research Institute, National Development and Reform Commission, Beijing 100038, China; 3Guangzhou Institute of Energy Conversion, Chinese Academy of Sciences, Guangzhou 510640, China

**Keywords:** WRF-Chem, carbon neutral, public health, air quality, Guangdong Province

## Abstract

In order to mitigate global warming and improve air quality, the transformation of regional energy structures is the most important development pathway. China, as a major global consumer of fossil fuels, will face great pressure in this regard. Aiming toward achieving the global 2 °C warming target in China, this study takes one of the most developed regions of China, Guangdong Province, as the research area in order to explore a future development pathway and potential air quality attainment until 2050, by developing two energy structure scenarios (BAU_Energy and 2Deg_Energy) and three end-of-pipe scenarios (NFC, CLE, and MTFR), and simulating future air quality and related health impacts for the different scenarios using the WRF-Chem model. The results show that under the energy transformation scenario, total energy consumption in Guangdong rises from 296 Mtce (million tons of coal equivalent) in 2015 to 329 Mtce in 2050, with electricity and clean energy accounting for 45% and 35%. In 2050, the transformation of the energy structure leads to 64%, 75%, and 46% reductions in the emissions of CO_2_, NOx, and SO_2_ compared with those in 2015. Together with the most stringent end-of-pipe control measures, the emissions of VOCs and primary PM_2.5_ are effectively reduced by 66% and 78%. The annual average PM_2.5_ and MDA8 (daily maximum 8 h O_3_) concentrations in Guangdong are 33.8 and 85.9 μg/m^3^ in 2015, with 63.4 thousand premature deaths (95% CI: 57.1–70.8) due to environmental exposure. Under the baseline scenario, no improvement is gained in air quality or public health by 2050. In contrast, the PM_2.5_ and MDA8 concentrations decline to 21.7 and 75.5 μg/m^3^ under the scenario with energy structure transformation, and total premature deaths are reduced to 35.5 thousand (31.9–39.5). When further combined with the most stringent end-of-pipe control measures, the PM_2.5_ concentrations decrease to 16.5 μg/m^3^, but there is no significant improvement for ozone, with premature deaths declining to 20.6 thousand (18.5–23.0). This study demonstrates that the transformation of energy structure toward climate goals could be effective in mitigating air pollution in Guangdong and would bring significant health benefits. Compared with the end-of-pipe control policies, transformation of the energy structure is a more effective way to improve regional air quality in the long term, and synergistic promotion of both is crucial for regional development.

## 1. Introduction

With the rapid development of urbanization and industrialization, China has become a major consumer of fossil fuels and the world’s largest carbon emitter in recent years [1,2,3]. To alleviate global warming and protect the atmospheric environment, the Chinese government signed the Paris Agreement [4] and committed to achieving carbon neutrality before 2030 and carbon peaking before 2060. At the same time, China is also suffering from significant environmental problems represented by high concentrations of PM_2.5_ (fine particulate matter 2.5 microns or less in diameter) and ozone due to rapid industrialization, urbanization, and increasing numbers of vehicles [5,6,7]. Long-term exposure to high levels of PM_2.5_ and ozone can harm human health, not only inducing cardiopulmonary disorders and impairments, but also contributing to a variety of other adverse health effects [8]. According to monitoring data released by the China National Environmental Monitoring Center, the PM_2.5_ concentration in China has frequently exceeded the national standard of 35 μg/m^3^ in recent years, especially the North China Plain and eastern coastal areas, where concentrations are more than two times higher [5,9], posing a severe threat to public health [10]. In the short term, the reduction in pollutant emissions can be achieved by strengthening end-of-pipe control measures, thus improving regional air quality. However, the effectiveness of end-of-pipe controls is limited by the level of technology and policy support in the long term, whereas the transformation of the social energy consumption structure is the fundamental way to improve air quality, which is a win–win strategy to promote the simultaneous reduction in greenhouse gases and atmospheric reactive pollutants [11,12,13]. Posing the significance of carbon neutrality and air quality improvement, in order to achieve sustainable development and guide policy making, it is important for China, as well as other countries and regions, to design multiple potential development pathways to adjust the energy structure of societies, as well as integrate with assessments of potential air quality and public health impact to quantify the synergistic benefits.

Guangdong Province consists of 9 major cities in the Pearl River Delta (PRD) region and 12 less-developed cities in the non-PRD. Guangdong is the most prosperous and populous region in southern China; the regional location and division of Guangdong are presented in Appendix A. The PRD region, which is known as “the world’s factory” and encompasses a wide range of industries, is one of the most developed regions in China and has accounted for 80% of the province’s total economy for over 10 years (Guangdong Statistics Yearbook). From 1978 to 2018, the total GDP of Guangdong grew 483 times, and Guangdong has gathered 8% of the total population and created nearly 11% of the total GDP with less than 2% of the national territory since the reform and opening [14]. Although the per capita income and urbanization rate are both in the leading position in China, the energy structure of Guangdong is still dominated by fossil energy, which has led to the extensive emission of air pollutants [15]. The provincial government has made great efforts to improve air quality, making Guangdong the first province to reach the national standard for PM_2.5_ concentration (35 μg/m^3^) in 2015. However, it still far exceeds the WHO’s PM_2.5_ guideline of 10 µg/m^3^ [16]. In addition, due to its special climate conditions [17] and negligence of emission controls for volatile organic compounds (VOCs) [15,18], Guangdong has also faced severe ozone pollution in the past few years [19,20]. Therefore, under the background of mitigating global warming, it is essential to explore a suitable development pathway in Guangdong to achieve transformation of the energy structure, and to synergistically reduce the emission of greenhouse gases as well as air pollutants, which will serve as a model for the development of other regions in China.

To guide future development planning in Guangdong as well as other regions in China, this study aims to explore possible development pathways for Guangdong toward changing future social energy structure to comply with the global 2-degree warming target, and to evaluate the potential synergistic benefits for air quality and public health. The structure of this study is introduced as follows: Section 2 is the literature review. Section 3 introduces the scenario design and the models (models used for scenario design, air quality simulation, health impact assessment) used in this study. Section 4 introduces the main results of social development under different scenarios in the future, including the change in social energy structure (Section 4.1), atmospheric pollutant emissions (Section 4.2), the change of atmospheric concentration of PM_2.5_ and ozone (Section 4.3) and the related public health impact (Section 4.4). Section 5 discussed the limitations and the main results of the study, and compared the present results with other studies. Section 6 provides some concluding remarks and research extensions.

## 2. Literature Review

A series of studies have explored future development pathways, as well as estimated future emissions and air quality in China by establishing different development scenarios. The area scope of the studies covers national [13,21,22,23,24,25,26,27], provincial [9,28,29], and city scales [30,31,32,33,34].

The main content of these studies could be grouped into three categories. The first category focuses on evaluating the change of air quality in China from a long-term perspective (from 2010 to 2035–2060), these studies mainly adopted the data provided by international widely-used scenario models due to their complete long-term scenario database, represented by the Greenhouse Gas and Air Pollution Interactions and Synergies (GAINS) model. For example, Xie et al. [27] used the GAINS model to predict the greenhouse gas and air pollutant emissions in China till 2050 by constructing two development scenarios, and combined with a global chemistry transport model and a health assessment model to estimate the health impacts and economic cost of PM_2.5_ pollution. Westervelt et al. [26] adopted two scenarios provided by the GAINS model to evaluate the impact of human pollutant emissions and climate change on the ozone concentration and health burden in China in 2050, respectively. Meanwhile, the new emission scenario developed by the Dynamic Projection model for Emissions in China (DPEC) [35] has also been adopted in studies in recent years. Xu et al. [33] used the emission scenario provided by the DPEC to evaluate the impacts of regional emission reduction towards carbon neutrality on air quality in 2060 in China. Chang et al. [34] and Zhang et al. [29] used DPEC emission scenarios to evaluate the improvement of PM_2.5_ pollution under the carbon emission mitigation pathways in Guangdong–Hong Kong–Macao Greater Bay Area (GBA) and Sichuan Provinces in 2035 in China, respectively.

The second category of studies focuses on medium-term (from 2010 to 2025–2030) development planning. The regional scope of these studies varied from the national to the city level. These studies tend to have a more specific description of the current situation and the development of the region, they have sufficient localized data (activity level, emission factor and so on) and are flexible in designing scenarios based on the study issues. For example, Yang et al. [13] estimated the impact of the popularity of the carbon emission trading system and electrification in the power and transportation sector on air quality in 2030 in China. Wu et al. [30] used Guangzhou city as a case to explore the air quality and public health co-benefit under three scenarios with different carbon reduction strengths, and highlighted the importance of promoting peaking carbon dioxide emissions for the improvement of air quality and public health at the city level. Jiang et al. [31] toke Shenzhen city as an example, and combined the Long-range Energy Alternatives Planning (LEAP) model with the WRF-CMAQ model to assess the impact of transdepartment and transregional synergic governance on Shenzhen’s greenhouse gas emission reduction and air quality improvement in 2030.

The third category of studies focuses on research from a sectoral perspective. For example, Peng et al. [36] evaluated the potential air quality and health benefits of different fossil fuel mitigation strategies for the industrial, power, residential, and transportation sectors in China, respectively, and further evaluated the benefits of the electrification of the transportation and residential sectors in 2030 [37]. Wang et al. [38] evaluated the environmental co-benefits of energy efficiency improvement in the coal-fired power sector by comparing three different development scenarios, whereas Zhang et al. [39] focused on evaluating the cement industry in China.

Although many studies have been conducted, as mentioned above, most of the studies still focus on the national and city levels, whereas the studies for the provincial scale are still insufficient [9,29], especially for southern China. In addition, existing studies at the provincial scale did not take climate target into consideration, but mainly evaluated the atmospheric co-benefit of respective carbon peaking pathways, and only focused on medium-term studies.

Therefore, to fill the research gaps mentioned above, with the achievement of the global two-degree warming target as a constraint, this study would focus on long-term development planning on the provincial scale, and choose a typical developed area—in this instance, Guangdong Province in southern China—to study.

## 3. Materials and Methods

### 3.1. Scenario Design

In this study, two energy scenarios and three end-of-pipe scenarios were developed to explore the emission reduction potential of energy structure adjustments and end-of-pipe control measures in Guangdong. 

The first energy scenario in this study was developed based on the Integrated Policy Model for China (IPAC) model. The IPAC model is a comprehensive policy evaluation model developed by the Energy Research Institute (ERI), which is a technology-based, bottom-up integrated assessment model to analyze global, national, and regional energy and environmental policies, and it has been widely applied to policy evaluation of energy and climate change in China [40,41,42,43,44]. The IPAC model has a variety of model approaches (http://www.ipac-model.org.c, (accessed on 20 August 2022)); the IPAC-AIM/technology model is a major component of the IPAC model, used to provide a detailed description of the current and future development of energy services and their equipment to simulate future energy consumption. It focuses on calculating the sub-variety energy demand for each end-use power sector under various future scenarios. To comply with the two-degree global warming target, this study uses the IPAC-AIM/technology model to quantitatively analyze the future energy transformation by constructing a low-carbon scenario for the future socio-economic development of Guangdong, which is called the 2Deg_Energy scenario. The key parameters in the model for the future social development of Guangdong are listed in Appendix A, and the major assumptions of social development of the model were introduced in the study of Jiang et al. [45], which were also summarized into a few points as follows:The industrial sector of Guangdong would be dominated by advanced manufacturing industries, leading to a continuous increase in electricity consumption. Meanwhile, Guangdong would undergo an economic structural transformation, and the service industry would take up a large proportion of the total economic volume, so the increase in industrial energy consumption would be limited, and the adjustment of energy structure would be the most important feature for the industrial sector.The transportation sector of Guangdong would widely adopt electricity in the future. Small electric vehicles would be the major transport tools for travel, whereas large trucks and ships would be gradually electrified. In addition, the volume of air transportation in Guangdong would grow further, leading to an increase in aviation kerosene use in the short term. The electric aircraft would start to become popular after 2030, and the total fuel use in the transportation sector would change smoothly until 2025, after which it would start to decline rapidly.For the residential sector in Guangdong, due to population growth and continuous urban expansion, as well as the rapid development of the service industry, the future energy consumption of buildings in Guangdong would continue to grow and be dominated by electricity. However, along with the continuous improvement of building energy use efficiencies, such as the development of advanced lighting, air conditioning and heating technologies, the future trend of energy use in the residential sector would remain flat.The power sector of Guangdong would widely adopt clean energy for electricity generation. With the completion of large-scale centralized wind power and nuclear power plants in the coastal areas of western and eastern Guangdong, the province’s electricity generation will rise to 705 billion kWh in 2050, an increase of 67% compared to 2015, whereas the share of electricity generation from traditional fossil fuels will be less than 5%, which is almost clean. However, due to population growth and urban expansion, the province’s electricity demand would reach 1 trillion kWh in 2050, and the additional demand will be met by green power imported from regions outside the province.

In this study, the energy consumption calculated by the IPAC model for different sectors as well as for different species are substituted into the Greenhouse Gas and Air Pollution Interactions and Synergies (GAINS) model to calculate the related emission of air pollutants. The second energy scenario is provided by the GAINS-Guangdong model. GAINS-Guangdong is a regional part of the GAINS model. The GAINS model, developed by the International Institute for Applied Systems Analysis (IIASA), is a comprehensive evaluation model for assessing the potential cost and environmental benefits of synergistic greenhouse gas and air pollutant emission reductions. It has been widely used in the accounting of regional air pollutant emissions and design abatement strategies [46]. Detailed descriptions of GAINS-China can be found on the website (https://previous.iiasa.ac.at/web/home/research/researchPrograms/air/GAINS.html, (accessed on 6 July 2021)). The GAINS-Guangdong model provides a baseline scenario for the development of the energy structure of Guangdong Province from 2015 to 2050, which is called the BAU_Energy scenario in this study. This scenario is established based on the prediction of the World Energy Outlook 2011 (WEO 2011), published by the International Energy Agency (IEA). In the assumption of this scenario, the energy structure of Guangdong Province will be dominated by traditional fossil fuel energy, representing the future development of Guangdong without any energy transformation measures. 

The three end-of-pipe control scenarios adopted in this study are provided by the GAINS-Guangdong model as well. The GAINS model provides three different end-of-pipe emission control scenarios with different technology penetration rates for each region of China, as introduced by Stohl et al. [47]. The Current Legislation (CLE) scenario, which is established by referencing the assessment of the 12th Five-Year Plan legislation discussed by Zhao et al. [48] and Wang et al. [49], is used in this study as a baseline scenario for end-of-pipe emission control development. The second scenario is the No-Further-Control (NFC) scenario, which assumes that the strength of future end-of-pipe emission control measures will remain at the same level as in 2010–2015; this represents the minimum emission reduction potential. The third scenario is the Maximum Technically Feasible Reduction (MTFR) scenario, which assumes the implementation of the best available end-of-pipe control measures, ignoring political or economic constraints but considering technical limitations; it reflects the maximum potential emission removal using the technical level of China’s 12th Five-Year Plan. 

Four combined scenarios are further established by combining the energy structure and end-of-pipe control scenarios mentioned above; these are listed in Table 1. The BAU scenario is the baseline scenario, representing the general future development pathway of Guangdong under the impact of announced policies. The 2Deg scenario represents the energy transformation pathway of Guangdong in which end-of-pipe control measures are the same as in the BAU scenario; differences in the results of the BAU and 2Deg scenarios will quantify the potential co-benefits of energy structure transformation in Guangdong. While the BAU_NFC scenario represents the worst situation for the development pathway in Guangdong, with a high share of fossil fuel energy and stagnant emission control measures, in contrast, the 2Deg_MTFR scenario represents the most expected development pathway for Guangdong, with low-carbon energy structure and the most stringent end-of-pipe control measures. A comparison of the BAU and BAU_NFC scenarios will indicate future improvement in air quality in Guangdong due to current pollution control legislation. A comparison of the BAU and BAU_NFC scenarios makes it possible to analyze the maximum potential of air quality improvement in Guangdong under the current level of legislation and emission abatement technology. To be more in line with actual conditions, this study updated the energy consumption data and other relative activity data in 2015 for the BAU_Energy and 2Deg_Energy scenarios based on the Guangdong Statistics Yearbook, China Energy Statistics Yearbook, and China Industrial Economy Statistical Yearbook to improve the reliability of the scenarios, and also to adjust future results in line with the changes.

### 3.2. Air Quality Simulation

This study uses the WRF-Chem v3.9.1 model to simulate changes in air quality under different scenarios. The WRF-Chem model is a regional air pollution model that can simulate the transport, mixing, and chemical transformation of fine particulate matter (PM_2.5_) and ozone (O_3_). A detailed description of the model is summarized by Grell et al. [50]. In this study, the WRF-Chem model uses 6 × 6 km^2^ horizontal resolution with 40 vertical levels from the surface to 50 hPa, and the center of the study domain is located at 113.5E and 23N, with grid numbers for 149 (WE) × 104 (SN). The SAPRC99 chemical mechanism [51] for gas-phase chemistry is used in this study, and the aerosol module is the Model for Simulating Aerosol Interactions and Chemistry (MOSAIC), which uses 4 volatility bins with the Volatility Basis Set (VBS) for organic aerosol evolution [52,53]. 

Major inputs to the model include meteorological fields, chemical initial boundary conditions, and emissions inventories. The meteorological fields are based on the National Centers for Environmental Prediction (NCEP) FNL data. The chemical initial and boundary conditions are provided by the global chemistry transport model MOZART-4 [54]. The study follows the temporal and spatial patterns of the Multiresolution Emission Inventory for China (MEIC, http://www.meicmodel.org/, (accessed on 25 June 2022)) to allocate the emission inventories calculated by GAINS for 2015 and 2050 into 6 × 6 km^2^ grids in the research domain. Since emission data from outside Guangdong are not available in the GAINS-Guangdong model, this study adopts the emission inventory provided by MEIC for regions outside Guangdong. Further details of the model configuration are provided in Appendix A. Considering the limitations of computing resources and the research content, the BAU, 2Deg, and 2Deg_MTFR scenarios are chosen to simulate changes in air quality in 2050, in addition to the simulation for the reference year 2015 for comparison. Meanwhile, January, April, July, and October are selected as four typical months for the simulations in this study, and the average PM_2.5_ and MDA8 concentrations in these four months are used to represent the annual average concentrations in Guangdong. Verification of the model performance is provided in Appendix A.

### 3.3. Health Benefits Assessment

In this study, the number of premature deaths (Mor) associated with long-term exposure to PM_2.5_ and O_3_ is calculated with the following equation:(1)$$Mor=\underset{d}{\sum }Pop\times y_{d}\times \left(1-\frac{1}{RR_{d}}\right)$$
where $y_{d}$ is the disease-specific baseline mortality rate in Guangdong, which is obtained from the World Health Organization (http://www.who.int/healthinfo/statistics/mortality_rawdata/en/, (accessed on 3 October 2021)). *Pop* is the exposed population of adults (>25 years old) in each grid cell, which is calculated from the 2015 gridded population data products provided by the Socio-Economic Data and Applications Center (SEDAC) [55] and the population age composition data from the Guangdong Statistical Yearbook. For the future population, this study adopts the population data predicted by GAINS and assumes that the spatial distribution of the population will remain the same as in 2015. $RR_{d}$ represents relative risk for a specific disease. Concentration-response (C-R) models represent the relationship between exposure and attributable mortalities. In this study, four diseases associated with long-term exposure to PM_2.5_ are considered, including chronic obstructive pulmonary disease (COPD), stroke, ischemic heart disease (IHD), and lung cancer (LC). The study applies the Integrated Exposure-Response model (IER) developed by Burnett et al. [56] to calculate $RR_{d}$ for PM_2.5_; the equation is as follows:(2)$$RR_{d}=\left\{\begin{array}{c} 1,\;C<C_{0}\\ 1+\alpha \times \left(1-e^{-\beta {\left(C-C_{0}\right)}^{\gamma }}\;\right),\;C\geq \;C_{0} \end{array}\right.$$
where C is the annual average PM_2.5_ exposure concentration for each grid simulated by WRF-Chem; $C_{0}$ is the threshold concentration below which no additional health risk is assumed; *α*, *β*, and *γ* are the parameters determining the overall shape of the C-R curves. The study adopts the IER parameter provided by Jiang et al. [57] for *α*, *β*, *γ*, and $C_{0}$. 

For health impacts related to long-term exposure to O_3_, the study focuses on the mortality risk from chronic obstructive pulmonary disease (COPD) and respiratory (RESP) diseases for adults. A log-linear relationship [58] is used to estimate premature mortalities attributable to O_3_ pollution:(3)$$RR_{d}=e^{\beta \left(C-C_{0}\right)}$$
where C is the annual average daily maximum 8 h O_3_ concentration (AMDA8) based on the WRF-Chem simulation; $C_{0}$ is the threshold concentration, set to 37.6 ppb, the same as in Yang, Zhao, Cao, and Nielsen [13]; *β* is the change in the certain death risk per ppb O_3_ increment, set to 0.0039 for COPD, which is provided by Liu et al. [59]; for RESP, *β* is calculated by the equation below [58]:(4)$$\beta =\frac{\ln\left(RR_{RESP}\right)}{\Delta C}$$
$RR_{RESP}$ represents the relative risk for RESP. This study adopts the data provided by Turner et al. [60], which sets $RR_{RESP}$ to 1.12 (95% CI: 1.08–1.16) for each 10-ppb increase in ozone concentration; therefore, *β* can be further calculated to be 0.0113.

Finally, based on the above calculations, the public health co-benefits (avoided premature deaths) due to air quality changes under the different scenarios can be obtained by comparing the differences in premature deaths.

## 4. Results

### 4.1. Energy Structure Comparison 

Figure 1 demonstrates the difference in future energy consumption under the two energy scenarios. In 2015, the total energy consumption of Guangdong is 296 Mtce (million tons of coal equivalent), and the industry, residential, transportation, and power sectors account for 34%, 13%, 22%, and 31% of the total energy consumption. From the perspective of energy type, traditional fossil energy (coal, oil products, and natural gas) accounts for 69% of total energy consumption and is the main source of energy consumption.

The total energy consumption of the BAU_Energy scenario reaches 710 Mtce in 2050, with a 140% increase compared with 2015. As can be seen in Figure 1, the energy consumption for all sectors shows an increasing trend until 2050, and the industry and power sectors have the most significant increments, 130 and 124 Mtce. For the industrial sector, the future increase in energy consumption comes mainly from the increase in clean energy, such as electricity consumption, which has an increment of 80 Mtce compared with 2015, followed by natural gas (31 Mtce). For the power sector, because it is still dominated by coal power, with the increase in the electricity load, the consumption of coal presents a significant increase of 74 Mtce compared with 2015. In contrast, nuclear and other renewable energy increase by 25 Mtce, accounting for 21% of the total increase. For the transportation sector, the consumption of traditional oil products increases dramatically due to the lack of electrification in the future, with a total increase of 86 Mtce, and total energy consumption reaches 154 Mtce, a 140% increment compared with 2015. For the residential sector, electricity consumption dominates the increase in the future, which reaches up to 67 Mtce in 2050. In parallel, the consumption of coal and biomass maintains a similar level as in 2015.

Under the 2Deg_Energy scenario, the transformation of the energy structure significantly decreases fossil energy consumption in Guangdong. Total energy consumption is 329 Mtce in 2050, with a reduction of 54% compared with the BAU_Energy scenario during the same period. The industrial, residential, transportation, and power sectors have a total consumption of 100, 45, 41, and 142 Mtce, respectively. Although the total energy consumption increases by 33 Mtce compared with 2015, the share of traditional fossil energy consumption decreases from 69% to 14%, being replaced by a significant increase in electricity and nuclear energy. The power and transportation sectors contribute the most to the reduction in fossil energy. For the power sector, although the total energy consumption shows an increment of 50 Mtce in 2050 compared with 2015, the increase comes mainly from the popularization of nuclear and renewable energy. Coal consumption decreases by 59 Mtce compared with 2015, and 134 Mtce compared with the BAU_Energy scenario in 2050. For the transportation sector, the total oil product consumption reduces by 8 Mtce in 2050, which accounts for only 19% of the total energy consumption of this sector. As can be seen in Figure 1, oil product consumption begins to significantly decrease after 2025, accompanied by a dramatic increase in electricity consumption due to the popularity of electric vehicles. For the residential sector, electricity dominates its energy structure in the future. In 2050, the total electricity consumption of the residential sector is 37.5 Mtce, accounting for 84% of total energy consumption. It should be mentioned that, compared with the BAU_Energy scenario, the electricity consumption of the residential sector is reduced by 29 Mtce in 2050, mainly due to the wide use of more efficient energy-saving technologies for household appliances. Finally, for the industrial sector, decarbonization is the most important feature; the reduction in coal and oil products in 2050 is 37 and 7 Mtce compared with 2015, with a 33 and 24 Mtce decrease compared with the same period in the BAU_Energy scenario. In contrast, electricity consumption accounts for 77% of the total energy consumption.

### 4.2. Emission Trends of Air Pollutant Emissions under Different Scenarios

Figure 2 demonstrates the emission trends for different pollutants under different scenarios in the future. 

For the emission of CO_2_, since end-of-pipe control measures aim to control atmospheric reactive pollutants, emissions are related only to the energy structure, which is the same for the BAU_NFC and BAU scenarios, and for the 2Deg and 2Deg_MTFR scenarios. As can be seen in the figure, total CO_2_ emissions are 579.4 MT (million tons) in Guangdong in 2015, with the industrial and power sectors dominating, accounting for 39% and 38% of total emissions. Under the baseline scenario (BAU_Energy for energy structure), the CO_2_ emissions increase up to 1038 MT in 2050, an increase of 44% compared with 2015. The increase in emissions originates mainly from the power and transportation sectors, which were 510 and 210 MT, respectively. Emissions from the industrial sector gradually decrease due to greater dependence on electricity consumption, but it is still one of the main contributors of CO_2_. Under the low-carbon development scenario (2Deg_Energy), emissions of CO_2_ continually drop down to 211 MT in 2050. The power sector contributes the most significant reduction, with 444 MT between the two development pathways in 2050, followed by the industrial (132 MT) and transportation sectors (147 MT).

As typical emission products of fossil fuel combustion, CO, NOx, and SO_2_ can be influenced by both energy structure and end-of-pipe control measures; the total emissions of these three pollutants are 6486, 1187, and 599 KT (kilo tons) in 2015, respectively. Under the BAU scenario, emissions of NOx and SO_2_ do not have a significant reduction in 2050, changing to 1186 and 903 KT, respectively, whereas CO dramatically reduces to 3159 KT. The emission of NOx decreases slightly before 2025 and then bounces back to 1186 KT in 2050, with the increase coming mainly from the transportation sector owing to the rapid increase in the number of fuel vehicles. In contrast, CO emissions continue to decline after 2015, especially before 2025. Comparing the BAU and BAU_NFC scenarios, emission control legislation makes a huge contribution to the reduction in NOx and CO in the transportation sector due to the popularity of new national standard vehicles and the phase-out of yellow label vehicles [61]. For SO_2_, it can be seen that the industrial and power sectors dominate emissions, which account for 71% and 22% in 2015, respectively, and emission control legislation leads to total avoided emissions of 192 KT in 2050, comparing the BAU and BAU_NFC scenarios; however, it still increases by 57 KT in the BAU scenario compared to the level in 2015. With the transformation of the energy structure, under the 2Deg scenario, total emissions of CO, NOx, and SO_2_ decline to 1984, 298, and 454 KT in 2050, a reduction of 69%, 75%, and 46% compared with the emission level in 2015. The transportation sector still dominates the reduction in the emissions of CO (959 KT) and NOx (521 KT) due to the popularity of electric vehicles. For the emission of SO_2_, the elimination of traditional high fossil fuel industries and coal-fired plants contributes most to the reduction. Together with the most stringent end-of-pipe control measures, under the 2Deg_MTFR scenario, the emissions of CO, NOx, and SO_2_ further reduce to 1040, 181, and 200 KT in 2050, respectively. Comparing the 2Deg and 2Deg_MTFR scenarios, the industrial sector makes the largest contribution to the reduction in the three pollutants, especially SO_2_, indicating that the industrial sector has a great potential to further reduce emissions of these three pollutants under the current level of pollutant abatement technology.

Unlike the above four pollutants, the emissions of primary PM_2.5_ and VOCs are closely related to daily life and industrial processes, which do not rely on the consumption of fossil fuels. Their total emissions are 393 and 1904 KT in 2015, respectively, with the industrial sector dominating. Under the BAU scenario, the total emissions of primary PM_2.5_ and VOCs decline slightly to 267 and 1717 KT in 2050. For primary PM_2.5_, the transportation and industrial sectors contribute the most reduction compared to the BAU_NFC scenario, with 54 and 42 KT, respectively. For VOCs, the popularity of new national standard vehicles makes a significant contribution to reduction, with 669 KT avoided in the transportation sector. However, the transformation of the energy structure cannot significantly mitigate the emissions of these two pollutants. The total emissions of primary PM_2.5_ and VOCs decline to only 194 and 1546 KT under the 2Deg scenario in 2050, a reduction of 10% and 27% compared with the BAU scenario during the same period. Emissions from the industrial sector dominate the rest of the emissions of primary PM_2.5_ (49%), because most emissions come from industrial processes, such as the production of brick and steel, which cannot be alleviated by reducing fossil energy consumption but instead rely on the implementation of stricter end-of-pipe controls. The same applies to the control of VOC emissions, which come mainly from the use and production of industrial solvents in industrial processes. Combined with the most stringent end-of-pipe control measures, under the 2Deg_MTFR scenario, the emissions of primary PM_2.5_ and VOCs further reduce to 87 and 645 KT in 2050, a reduction of 55% and 58% compared with the 2Deg scenario during the same period, mainly from the industrial sector. However, there is still a large number of emissions from the industrial sector of these two pollutants, especially VOCs, with the industrial sector accounting for 87% of total VOC emissions. The results reveal that, on the one hand, there was negligence in the planning of emission controls for industrial processes during the 12th Five-Year Plan in Guangdong, especially for the control of primary PM_2.5_ and VOCs. On the other hand, there is still considerable room for progress in industrial emission abatement technology and related control policies due to the large number of residual emissions under the 2Deg_MTFR scenario. In the future, the industrial sector will be the most important source for the reduction in primary PM_2.5_ and VOCs.

### 4.3. Air Quality Changes under the Different Scenarios

In this study, the BAU, 2Deg, and 2Deg_MTFR scenarios are selected for the simulation of air quality. A total of four simulations are conducted, corresponding to air quality in 2015 and 2050 under the three scenarios. In 2050, the comparison between the 2Deg and BAU scenarios represents the improvement in future air quality in Guangdong due to the transformation of the energy structure, whereas the comparison between the 2Deg and 2Deg_MTFR scenarios represents the improvement due to the strictest end-of-pipe emission control measures based on energy structure transformation.

#### 4.3.1. PM_2.5_


Figure 3 illustrates the spatial distribution of model-simulated PM_2.5_ concentration in 2015, and the corresponding spatial changes in 2050 under the three scenarios. 

In 2015, the annual average PM_2.5_ concentration in the 21 cities in Guangdong is 33.8 μg/m^3^, barely meeting the national secondary standard for annual average PM_2.5_ concentration (35 μg/m^3^). The annual average concentration in the nine cities in the Pearl River Delta (PRD) is 34.4 μg/m^3^, whereas for cites outside the PRD it is 33.40 μg/m^3^. Foshan, Guangzhou, Dongguan, and Zhaoqing in the PRD region have the most severe pollution, with 39.6, 38.7, 36.5, and 38.9 μg/m^3^, respectively. In addition, Chaozhou and Jieyang in the eastern part of Guangdong are also seriously polluted, with 38.7 and 38.9 μg/m^3^.

In 2050, there is no significant reduction in the concentration of PM_2.5_ under the BAU scenario, with an average concentration of 34.2 μg/m^3^ in the 21 cities. The concentration in Heyuan in northern Guangdong improves relatively significantly, with a decrease of 3.08 μg/m^3^, whereas cities in eastern Guangdong have an average decrease of 1.9 μg/m^3^. However, for central Guangdong, the concentration in cities such as Shenzhen, Zhongshan, and Zhuhai in the Pearl River Delta is exacerbated by 1.1, 1.4, and 1.5 μg/m^3^. 

Under the 2Deg scenario, the PM_2.5_ concentrations in all cities are significantly reduced, with a range of 8.8–16.9 μg/m^3^. The average concentration is 21.7 μg/m^3^, with a reduction of 12.1 μg/m^3^ compared with 2015. Guangzhou, Foshan, and Jiangmen, three typical cities with well-developed manufacturing industries in the PRD region, have the most significant reduction, with 16.9, 16.8, and 16.7 μg/m^3^.

If the strictest end-of-pipe control measures are implemented, under the 2Deg_MTFR scenario, the annual average PM_2.5_ concentration further declines to 16.5 μg/m^3^ in 2050, which is closer to the WHO guideline of 10 μg/m^3^. The improvement is the most obvious in the PRD region, with an average concentration of 15.9 μg/m^3^ for the nine cities, which have a decrease of 18.5 μg/m^3^ relative to 2015. A comparison of the cities shows that the pollution level among the 21 cities in Guangdong would be similar. Although significant improvements occur throughout the province under this scenario, there is still a gap of 6.4 μg/m^3^ compared with the WHO standard. On the one hand, this study does not consider synergistic emission reductions in the areas surrounding Guangdong, and the cross-regional transmission of those pollutants could potentially lead to an increase in local PM_2.5_ concentration in Guangdong. In addition, the emission reduction measures in this study are based on the legislation in the 12th Five-Year Plan, and there are policies and technologies that have not yet been considered compared with the new 14th Five-Year Plan. 

#### 4.3.2. Ozone

In terms of ozone concentration, this study analyzes the spatial distribution of the annual average daily maximum 8 h ozone concentration (MDA8) in Guangdong, as shown in Figure 4. 

In 2015, the ozone pollution in Guangdong presents different spatial characteristics, in which the cities in the central PRD region are heavily polluted, with the annual average MDA8 concentration reaching 100.6 μg/m^3^, as represented by Dongguan (106.6 μg/m^3^), Foshan (104.6 μg/m^3^), and Guangzhou (107.1 μg/m^3^), which have severe PM_2.5_ pollution at the same time. The annual average MDA8 concentration in the 21 cities in Guangdong is 97.4 μg/m^3^. 

Under the BAU scenario, the MDA8 concentration in Guangdong does not show any significant improvement in 2050, with an average decrease of 1.5 μg/m^3^ for cities in the PRD and 0.4 μg/m^3^ for cities outside the PRD. For the emissions of NOx and VOCs, which are the two major chemical precursors of ozone [62], as seen in Figure 2, total NOx emissions under this scenario in 2050 are the same as in 2015, whereas VOCs increase little, leading to continued ozone pollution.

After the transformation of the energy structure, the MDA8 concentration in Guangdong decreases substantially in 2050 under the 2Deg scenario. The annual average MDA8 in the 21 cities in Guangdong is 85.7 μg/m^3^, a reduction of 12% compared with 2015, with an average reduction of 15.2 μg/m^3^ in the PRD cities and 10.2 μg/m^3^ in the non-PRD cities. Dongguan, Foshan, Jiangmen, and Guangzhou in the PRD region have the most significant reduction, with 20.8, 17.3, 18.3, and 18.2 μg/m^3^, respectively. 

However, there is no further reduction under the 2Deg_MTFR scenario. In 2050, the annual average MDA8 concentration of Guangdong is 85.4 μg/m^3^, the same as in the 2Deg scenario. These results suggest that the technology level for emission controls in the 12th Five-Year Plan cannot further alleviate ozone pollution after the transformation of the energy structure. Because ozone is a typical secondary atmospheric pollutant, in contrast to PM_2.5_, the concentration of ozone is highly correlated with the nonlinear relationship between the emission of NOx and VOCs [62], and also with the background concentration [63]. The synergistic reduction in NOx and VOCs is the key to suppressing ozone intensification, which should be fully considered in technology updates and the legislation of future control measures.

### 4.4. Public Health Co-Benefits under the Different Scenarios

Figure 5 illustrates the cumulative distributions of the population under different exposure levels to PM_2.5_ and MDA8 by scenario, with the dashed line representing the concentration standard set by the WHO guidelines (10 μg/m^3^ for PM_2.5_ and 100 μg/m^3^ for MDA8). Under the BAU scenario, the exposure level is not significantly alleviated, with 39.9% and 36.7% of the population exposed to MDA8 concentrations above 100 μg/m^3^, and 98.6% and 98.5% exposed to PM_2.5_ above 10 μg/m^3^ in 2015 and 2050, respectively. With energy structure transformation, the population exposed to MDA8 in 2050 is significantly reduced, with only 0.94% of the population exposed to concentrations above 100 μg/m^3^, mainly in the PRD region. In contrast, for PM_2.5_, the population exposed to more than 10 μg/m^3^ decreases from 98.5% to 84.9% over the same period, which needs to decrease further still compared with MDA8. By incorporating energy structure transformation and the strictest end-of-pipe control measures, the population exposed to PM_2.5_ concentration over 10 μg/m^3^ is further reduced to 31% in 2050, with a large improvement compared with the 2Deg scenario, but with little change in MDA8 exposure levels.

In 2015, the total premature mortality attributed to long-term exposure to PM_2.5_ and ozone is 63.4 thousand (95% CI: 51.7–70.8), which is similar to the results of Liu et al. [64]. The mortality from O_3_ is much lower than from PM_2.5_. For exposure to PM_2.5_, deaths from IHD, COPD, STK, and LC are 21.8 (19.8–24.1), 6.3 (5.6–7.5), 26.2 (23.6–28.7), and 5.2 (4.6–5.8) thousand, respectively. For ozone, deaths from RESP and COPD are 1.5 (1.4–1.8) and 2.4 (2.2–2.9) thousand (Figure 6). 

The total deaths related to PM_2.5_ and O_3_ reduce slightly in 2050 under the BAU scenario, by 0.9 thousand. With energy structure transformation, total premature deaths decrease to 35.5 (31.9–39.5) thousand under the 2Deg scenario, a reduction of 44% compared with 2015. The avoided deaths come mainly from STK induced by PM_2.5_, with a reduction of 14.8 thousand, followed by IHD (5.8 thousand). Finally, under the 2Deg_MTFR scenario, premature deaths further decrease to 20.6 (18.5–23.0) thousand, a reduction of 68% compared with 2015, attributed mainly to reductions in STK (22.4 thousand) and IHD (10.3 thousand) induced by PM_2.5_. 

## 5. Discussion

In this study, four scenarios for future development in Guangdong are developed by adopting different energy structure adjustments and end-of-pipe control assumptions. For the BAU scenario, which represents the baseline development pathway for Guangdong, the energy structure remains traditional fossil energy intensive. Although the continued implementation of end-of-pipe control measures can effectively reduce pollutant emissions, over-reliance on fossil energy still hinders the improvement of air quality in 2050, which remains at the same level as in 2015, this result is lower than the study of Tang et al. [25], in which the annual average PM_2.5_ concentration in China decreased in the range from 8% to 12% in 2050 compared with 2015 under the Shared Socioeconomic Pathways Level1 (SSP1, represent sustainability pathway [21]), which is expected due to the consideration of climate target in SSP1. The comparison of the BAU and BAU_NFC scenarios reveals that the emission control legislation in the 12th Five-Year Plan in Guangdong is effective in avoiding increases in air pollutant emissions, particularly for CO, VOCs, and NOx, which play a critical role in maintaining air quality under the development pathway with a high share of fossil energy. Li et al. [27] found that the emission control legislation could contribute to a total avoidance of nearly 7 Mt of SO_2_, 16Mt of NOx and 2 Mt of primary PM_2.5_ emissions in 2050 in China, respectively, and could reduce nearly 0.15 million premature deaths; this result is in accordance with this study. However, this only maintains air quality without deterioration, and there is no further improvement in the future. Transformation of the energy structure is an important step toward improved air quality in 2050, with significant decreases in the emission of pollutants closely related to fossil fuel consumption under the 2Deg scenario, such as CO_2_, NOx, and SO_2_, which contribute to significant reductions in atmospheric PM_2.5_ and MDA8 concentrations of 35% and 12% compared with 2015, whereas premature deaths are reduced by 44%. Compared with the study of Xie et al. [65] which focused on achieving the two-degree target on a national scale, the PM_2.5_ concentration in Guangdong would reduce nearly 20 μg/m^3^—the value greater than 12.1 μg/m^3^ in this study. This might be due to the much coarser grid resolution (2.5° × 1.9°) of Xie et al. [65] compared with this study, and also to the uncertainty of the air quality model. Merely relying on the transformation of the energy structure cannot further effectively reduce the emissions of VOCs and primary PM_2.5_, as these pollutants are emitted mainly from non-energy processes that depend on control from end-of-pipe reduction measures. This result also discloses the negligence of the emission controls for industrial processes in the legislation of the 12th Five-Year Plan. As discussed in Bian et al. [15], the control measures enacted during the 12th Five-Year Plan were effective in reducing SO_2_ (23%), NOx (11%) and CO (7%) emissions in Guangdong from 2010 to 2015; however, the emission of VOC adversely increased by 17% during the same periods, making it urgent to improve the control over VOC emissions in the future. If the most stringent end-of-pipe measures are further adopted based on the transformation of the energy structure, the emissions of VOCs and primary PM_2.5_ decline significantly in 2050, with the concentration of atmospheric PM_2.5_ concentrations decreasing by 51% relative to 2015 and premature deaths reducing by 68%; however, little improvement is obtained for ozone compared with the 2Deg scenario during the same period. Compared with the study of Liu et al. [64], which combined the end-of-pipe control scenario MTFR_Contr and the energy structure scenario BAU_Energy defined in this study to simulate changing PM_2.5_ concentration and health benefits in 2050, the results showed that the concentration of PM_2.5_ declined by 8.3 μg/m^3^ in the PRD region compared with 2015. This is smaller than the decrease of 18.5 μg/m^3^ and 13.2 μg/m^3^ for the combined 2Deg_MTFR and 2Deg scenarios in this study, indicating that the energy structure transformation in Guangdong plays a more critical role in improving air quality than merely strengthening end-of-pipe controls.

Meanwhile, this study has some uncertainties and limitations. First, the projection of future social activities and emission controls is complicated due to incomplete knowledge of future polices, technology development, and so on. In this study, the experiments were conducted based on macro assumptions for Guangdong in the future, and more elaborate consideration is needed for various processes and pollution sources. In addition, the study adopted end-of-pipe control measures referring to the technical level and legislation of the 12th Five-Year Plan, although these are still widely used in recent studies [21,26,66,67] due to the complete parameter set; however, parts of these are still inconsistent with China’s up-to-date emission control planning. Second, changes in emissions beyond Guangdong were not considered and evaluated in this work, and air quality could possibly be influenced by neighboring provinces or other countries [68] in the simulation results. Third, the simulation of pollutant concentrations in the WRF-Chem model could be affected by meteorological conditions, such as wind speed, wind direction, temperature, and wet deposition from precipitation, which vary in different seasons, locations, and years. The meteorological fields in the WRF-Chem model remained the same as in 2015 for the base year and the three future scenarios; therefore, the impacts of future meteorological conditions on PM_2.5_ and ozone have been ignored, which could affect the accuracy of the results [69,70]. Finally, uncertainty exists in the assessment of health impacts. The IER model adopted in this study could introduce uncertainties arising from the model itself and the assumptions on which it is based, which have been statistically estimated by a simulation approach described in Burnett et al. [56]. Meanwhile, the widely used C-R models were generally developed on the basis of epidemiological studies in the US and Europe and might cause bias when they are used in China. Moreover, the study assumed that there would be no change in the spatial distribution of the population and baseline mortality rate in the future and ignored the population aging problem. Such assumptions might lead to a possible underestimation of health benefits due to the higher risk of elderly people.

## 6. Conclusions

Aiming toward the global two-degree warming target, this study explores the future development pathway of Guangdong Province by constructing four integrated development scenarios (BAU_NFR, BAU, 2Deg, 2Deg_MTFR), based on two energy scenarios (BAU_Energy and 2Deg_Energy) and three end-of-pipe control scenarios (NFC, BAU, MTFR), and quantifying their effects on air quality as well as public health.

The results indicate that, with the goal of achieving the global two-degree warming target, transformation of the energy structure can effectively reduce greenhouse gas emissions while significantly improving regional air quality and reducing the number of premature deaths as compared to maintaining a traditional energy structure with a high share of fossil fuels. Furthermore, this study also demonstrates that, compared with end-of-pipe control policies, energy structure transformation will be more significant for regional air quality improvement from a long-term perspective. However, energy structure transformation alone still cannot achieve the WHO guidelines for air quality standards in Guangdong. For future air quality improvement, it is necessary to focus on the synergistic adjustment of the energy structure and end-of-pipe control measures, as well as updating policies for pollution control and related emission abatement technologies.

Meanwhile, it should be noted that, the current studies (including this study) have conducted research focusing on a single region. Since curbing global warming is a common goal that needs joint efforts among different countries, provinces, or cities, it is necessary to conduct studies that focus on the synergistic development between regions in the future.

## Figures and Tables

**Figure 1 ijerph-19-14965-f001:**
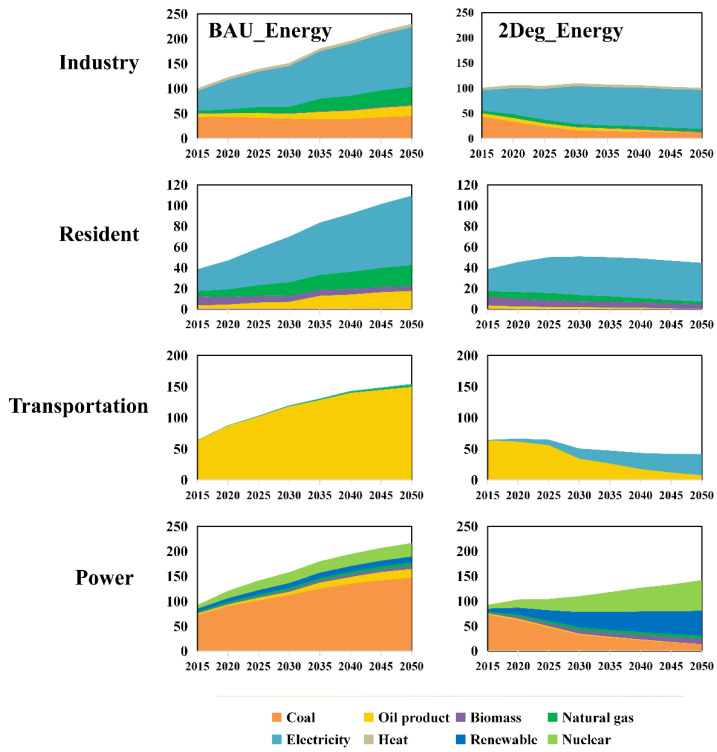
Comparison of energy structure change between the BAU_Energy and 2Deg_Energy scenarios in the future, units: Mtce.

**Figure 2 ijerph-19-14965-f002:**
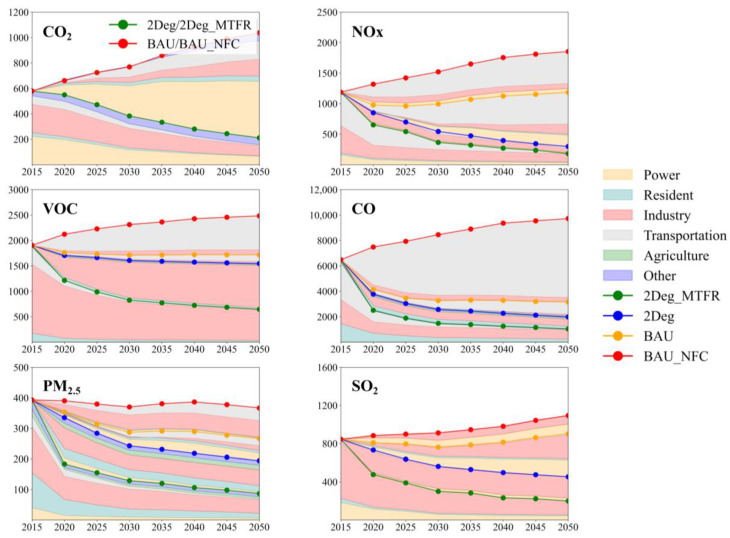
Emission trends of different pollutants under the different scenarios. For CO_2_ emissions, the unit is MT (million tons), whereas for the other pollutants the unit is KT (kilotons).

**Figure 3 ijerph-19-14965-f003:**
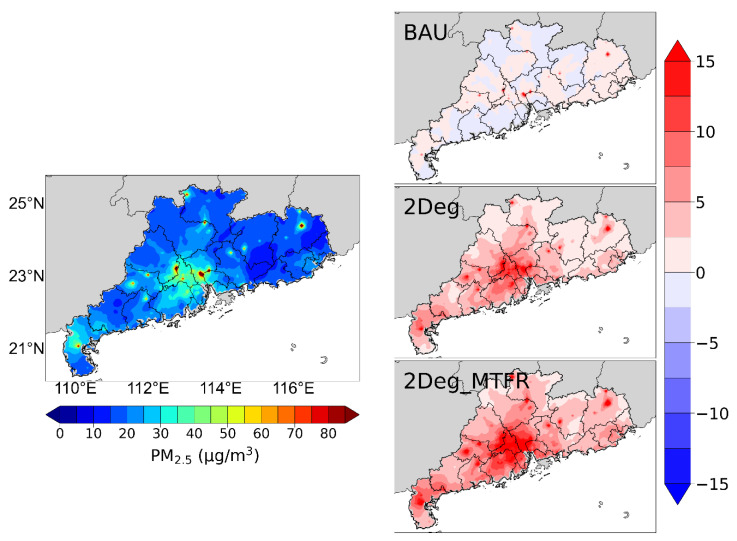
Spatial distribution of PM_2.5_ concentrations in Guangdong in 2015 (**left figure**). The changing PM_2.5_ concentrations under the three scenarios in 2050 (**right figures**), with red representing decreasing concentrations relative to 2015, and vice versa for blue.

**Figure 4 ijerph-19-14965-f004:**
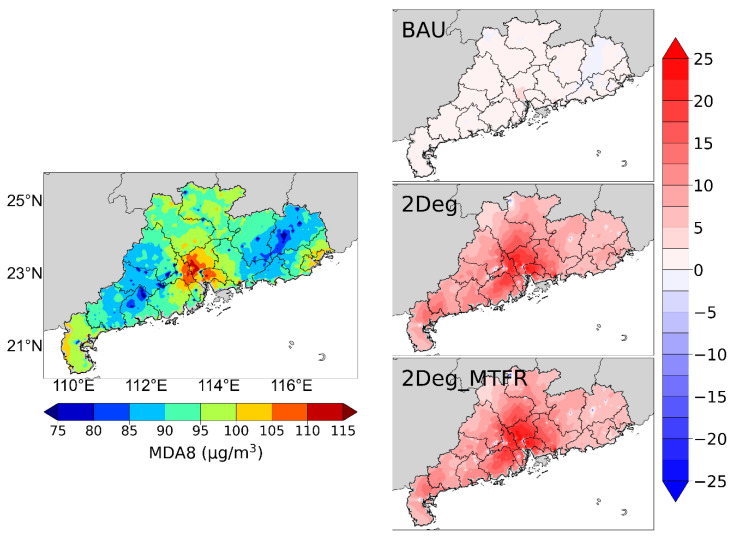
Spatial distribution of MDA8 concentrations in Guangdong in 2015 (**left figure**). The changing MDA8 concentrations under the three scenarios in 2050 (**right figures**), with red representing the decreasing concentrations relative to 2015, and vice versa for blue.

**Figure 5 ijerph-19-14965-f005:**
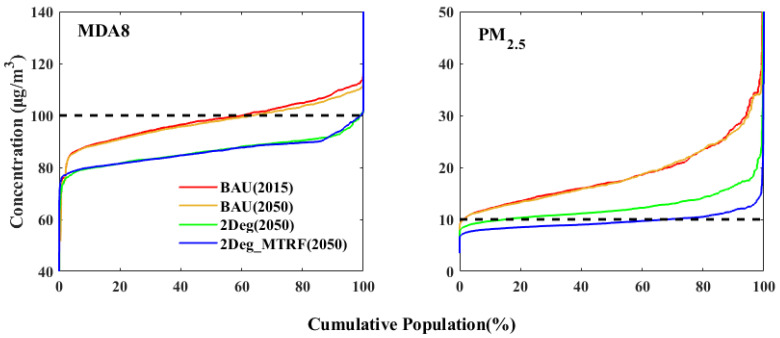
The cumulative distributions of annual mean PM_2.5_ and MDA8 exposures under the different scenarios.

**Figure 6 ijerph-19-14965-f006:**
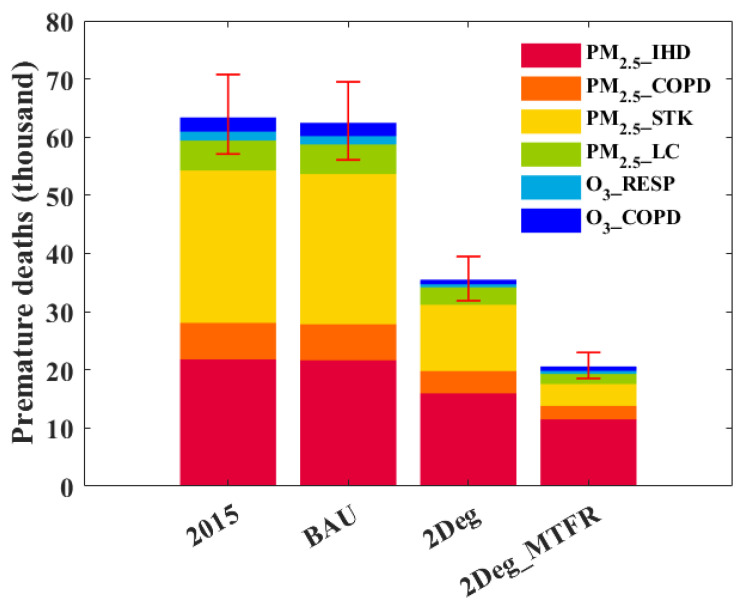
Premature deaths due to exposure to PM_2.5_ and ozone under the different scenarios.

**Table 1 ijerph-19-14965-t001:** Scenario design of the study.

Scenario	Energy Structure Scenario	End-Of-Pipe Control Scenario
BAU_NFC	BAU_Energy	NFC
BAU	BAU_Energy	CLE
2Deg	2Deg_Energy	CLE
2Deg_MTFR	2Deg_Energy	MTFR

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
