# Peer review of "Co-Benefits of Energy Structure Transformation and Pollution Control for Air Quality and Public Health until 2050 in Guangdong, China"

_ijerph, 2022, doi:10.3390/ijerph192214965_

Round 1
Reviewer 1 Report
The article is very interesting. It emphasizes the dual effect of emission reduction. There are some serious problems in this article.
1. The Chinese government has clearly put forward the goal of carbon peaking (2030) and carbon neutralization (2060). There are mandatory goals that must be achieved. Anything that does not meet the carbon peak and carbon neutral scenarios would have no practical significance. But this article has not adopted and discussed.
2. The study assumed that there would be no change in the spatial distribution of the population. But in fact, the total population, structure and spatial distribution will change dramatically. Some Chinese scholars have given and shared the data of future population distribution, so it is recommended to adopt it.
https://cstr.cn/31253.11.sciencedb.01683
Author Response
Thank you, and please see the attachment.

Reviewer 2 Report
The paper is interesting and addresses the very important issue of mitigating global warming and improving air quality in the 2050 perspective. Based on the example of the most developed region in China, five scenarios are presented, two of which deal with energy structure (BAU_Energy and 2Deg_Energy) and three with end-of-pipe scenarios (NFC, CLE, and MTFR), which take into account the projection of future air quality and related health impacts. The presented study demonstrates that changing the energy mix is a critical component of meeting climate targets and improving air quality in the 2050 horizon.
In my opinion, this manuscript can be accepted for publication. Nevertheless, I request that the authors re-read the paper to eliminate a few typos.
Author Response
Thank you for your suggestion! We have checked the paper in detail for spelling and grammar, and eliminated the typos.
Reviewer 3 Report
Review Report
On
"Co-benefits of energy structure transformation and pollution control for air quality and public health until 2050 in Guangdong, China”
ID: ijerph-1934677
This paper explore a suitable development pathway in Guangdong (China) focuses in the design of its energy structure in order to achieve a mitigation of pollution and thus avoid premature deaths
Areas of strength
The topic of this paper is interesting. It is well-written and readers can easily understand this study. Abstract gives a very clear and wide explanation of this work.
Areas of weakness
The manuscript has some basic flaws that are shown below:
- The goal of this paper should be clearly specified in the introduction. Also the structure of the work should be inserted in the introduction.
- The literature review section is missing. In this section should be included research related to the topic of the work. The relevant works that have been used in the research should be specified in an orderly and systematic way.
- The results were widely explained. However, the discussion part was very limited. There is no clear discussion of the results of the current study with the past literature to expand the body of knowledge with the related research studies
- The content of the conclusions section should be rewritten. It is basically a summary of the results. The conclusion section should help to refocus the reader's attention to the most important points and supporting evidence of paper’s arguments or position that is presented in the research. Conclusions should serve as a basis for continuing research, creating new ideas to resolve an issue you highlighted in the paper or offering new approaches to a topic.
Overall, the paper has an interesting contribution to the readers but some issues should be revised
Author Response

(The authors gave the same response as above.)

Round 2
Reviewer 3 Report
The authors have satisfactorily followed all the suggestions made, so the paper could be published, if the editor considers it.